# Mental Health Problems and Needs among Transitional-Age Youth in Indonesia

**DOI:** 10.3390/ijerph18084046

**Published:** 2021-04-12

**Authors:** Fransiska Kaligis, Raden Irawati Ismail, Tjhin Wiguna, Sabarinah Prasetyo, Wresti Indriatmi, Hartono Gunardi, Veranita Pandia, Clarissa Cita Magdalena

**Affiliations:** 1Department of Psychiatry, Faculty of Medicine, Universitas Indonesia, Jakarta 10430, Indonesia; raden.irawati@ui.ac.id (R.I.I.); tjin.wiguna@ui.ac.id (T.W.); citamagdalena@gmail.com (C.C.M.); 2Faculty of Public Health, Universitas Indonesia, Depok 16424, West Java, Indonesia; sabarin1@ui.ac.id; 3Doctoral Program of Medical Science, Faculty of Medicine, Universitas Indonesia, Jakarta 10430, Indonesia; wresti.indriatmi@ui.ac.id; 4Department of Dermatovenereology, Faculty of Medicine, Universitas Indonesia, Jakarta 10430, Indonesia; 5Department of Child Health, Faculty of Medicine, Universitas Indonesia, Jakarta 10430, Indonesia; hartono.gunardi@ui.ac.id; 6Department of Psychiatry, Faculty of Medicine, Universitas Padjajaran, Bandung 40115, West Java, Indonesia; veranitapandia0107@gmail.com

**Keywords:** coping mechanisms, health service expectation, mental health needs, mental health problems, transitional-age youth, adolescence

## Abstract

“Transitional-age youth” describes those whose ages range from 16–24 years old. In this phase, the youth face new challenges and new experiences which may increase the risk for having mental health problems, yet not very many seek help from mental health professionals. In Indonesia, no data are available about mental health problems and the needs of transitional-age youth. This study explores common mental health problems experienced during this stage and assesses how they cope with problems and their expectations from health services. This was a cross-sectional study involving 393 Indonesians aged 16–24 years in May 2020. More than 90% of students had financial and academic difficulties and felt lonely. The most prevalent mental health problem among students was anxiety (95.4%). Most of the students, ranging from 90% to 96.4%, had positive coping strategies. However, around 50% of respondents reported self-harming and having suicidal thoughts. The results of subcategories analysis between ages and faculties were similar. Their most important expectations from mental health services included confidentiality (99.2%) and being welcoming and friendly (99.2%). In conclusion, this study highlighted the most common problems transitional-age youth experience in Indonesia. While some of them already knew how to deal with their problems, not all the participants had good coping mechanisms. Their healthcare expectations were also explored, thereby providing a useful background to revise and amend the current conditions.

## 1. Introduction

Transitional-age youth refers to those in their late adolescence but still cannot be categorized into adulthood. This term describes those whose ages range from 16–24 years old, moving from adolescence into adulthood. At this stage, they face new challenges, have new experiences, and tackle new environments without prior knowledge and most probably in the absence of proper guidance. Some of them might need to face this period while being away from home and parents. As they mature during this period, their personalities are shaped. They are also given more responsibility and expected to manage and cope with their problems. Moreover, adaptation skill is developed during this period to deal with the physical and psychological changes [1,2].

Adolescent itself is defined by WHO as people whose ages range from 10–19 years old, while youth refers to people in the 15–24 years age group. The United Nations then recognized the category of “young people” as those in the 10–24 years age group [3]. In measuring the global burden of disease in young people aged 10–24 years old, Gore et al. divided the age group into 10–14, 15–19, and 20–24 years groups to measure and compare morbidity including neuropsychiatric disorders in those groups [4]. Sawyer et al. [5] proposed that adolescence should be defined as the 10–24 years period of age due to the documented start of puberty as young as age 10 years and that neurocognitive and emotional maturation continues over the age of 20 years. By including the 20–24 years period, focus can be given to the critical transition period between secondary education to employment.

Psychologically, in this transitional phase, things could become challenging for a young person, and they could face difficulties as they develop, which could impact their mental health and wellbeing. The demands of independent living and adjusting to many changes in life, for example, living on their own, managing own expenses, and adapting to new study or workplace could be the precipitating factors of mental health problems during transitional age. New relations, friendships or intimate relationships, are also formed. Their behavior and way of thinking also change as a response to this new environment. Along with all of this growth and transformation comes a threat for those young adults. When one has difficulty adapting to this transition phase, one is at risk of having mental and physical health issues [2]. Resilience could be a significant protective factor. People with good resilience tend to have adaptive coping mechanisms when faced with challenges and problems. Good resilience can help people recover from setbacks and adapt appropriately, using the challenges they have faced as learning tools to help them deal with future issues.

On the other hand, poor resilience can lead to long ruminations, feeling overwhelmed, and turning to unhealthy coping mechanisms. It might also lead to drug or alcohol abuse, mental health issues, and self-harm. Alcohol and substance abuse could be turned to as maladaptive mechanisms to escape problems. Mental disorders that may develop include depression, bipolar disorder, anxiety, schizophrenia, and other psychoses [6]. According to a study by Kessler et al. (2007), mental illnesses ranging from psychosis to substance-use disorder commonly occurred during the early 20’s. This is supported by the findings from studies done by Amminger et al. (2006) in Australia which found the median age for the presentation of first psychosis was 22, and Lauronen et al. (2007) in Finland which found the median age of onset for schizophrenia was 23. Psychotic problems such as schizophrenia usually occur between 15–17 years old. However, they typically receive treatment in the later years [7,8,9]. Having mental illness in the transition period can interfere with proper development, as significant development of cognitive, social, and emotional functions happens during this phase, mostly between 18–24 years old [10].

Unfortunately, when mental health problems arise during this transition, youth might not have the skill to manage their health which make them feel hesitant to ask for help. This could be due to low mental health literacy and the expectation from society for the young people to be fully responsible of their own life. This way of thinking prevents them to access counseling assistance or mental health service because they may be afraid to be considered incapable of dealing with their problems. Despite the frequency of suicidality, anxiety, and depression in this population, studies have shown that 60–80% of transitional-age youth are not in treatment [11,12]. Stigma related to mental illness makes it harder to receive the necessary support, even though people with mental health problems need social and emotional support from their surroundings and their closest peers. Negative stigma results from a lack of awareness and education about mental health [13].

According to the World Health Organization (WHO), one out of four people have mental illness. Every 40 s, one dies from suicide worldwide. The burden of mental illness is growing and notably impacting the economy and social structures negatively. Moreover, the cost of the treatment is considerably high, which contributes to the burden. In developing countries, 76–85% of low- and middle-income citizens do not receive proper treatment [14].

In Indonesia, healthcare system coverage for mental health needs to be evaluated. Unfortunately, mental health issues were not prioritized until recently, in contrast to other countries where they have established universal health coverage (UHC) schemes with mental health issues included, such as Australia and the UK [15]. In 2014, “pasung“ or physical restraints for those who have mental illness drew a lot of national and international attention, which created pressure for the national legislature, to pass the Law on Mental Health. The law included the mandate for community-based health services to provide access to mental illness. Furthermore, the government also attempted to raise awareness by implementing several mental health associated activities, training staff, and health professionals in more than 6000 community health centers [16].

Considering the high prevalence of mental health problems during the transitional-age youth and the current treatment gap, it is of utmost importance to take action to prevent and reduce those issues [17]. However, different approach should be provided compared to younger children or adulthood. University students represent transitional-age youth who are in developmentally challenging stage to adulthood. Untreated mental disorder during this period has an impact on academic performance, social relationships, and further mental health issues. Campuses should offer opportunities to address mental health support among late adolescents and young adults [10].

Among university students, those in medical schools have been studied exclusively and particularly those in Asia were found to experience common mental health problems such as depression, anxiety disorder, and suicidal ideation. Furthermore, unhealthy coping mechanisms such as consuming alcohol, smoking cigarettes, using internet for pornography, or cyberattacks were also found to be prevalent. Research undertaken by Zeng et al. discovered that the pooled prevalence of depression in Chinese medical students was higher than for students from other majors. This could be due to higher academic pressure in the medical education system, novel and complex learning materials compared to previous education, stricter rules, and demanding situations. Other possible reasons were longer study hours, lesser leisure time, and financial burden [18,19]. Unfortunately, medical students also faced several barriers in help-seeking. Students noted stigma, shame in admitting stress, or mental health problems such as weakness and fear of affecting their future career as doctors [20].

To give the right intervention, professional helpers need to know what mental health problems those in the transition period commonly experience and what they need. To the best of our knowledge, such data were not previously researched in Indonesia. This study explores mental health problems experienced by transitional-age youth, how they usually cope with their problems, and care they expect from mental health service.

## 2. Materials and Methods

### 2.1. Subjects

This was a cross-sectional study involving transitional-age youth aged 16–24 years in Indonesia. The inclusion criteria were university students, from any faculty, in Indonesia who gave consent to complete the survey and that the collected data can be used for research purposes. The sample was calculated using a single proportion formula with a 5% margin of errors and 95% confidence level, and estimation of the population was 100,000. Thus, we need 384 subjects to participate [21]. The data were collected in May 2020 through an online questionnaire. The survey was emailed to different university student associations represented three zones in Indonesia, which are west, center, and east. We asked them to spread among members in two weeks. Three hundred ninety-three students responded.

### 2.2. Instrument

Demographic data were collected, including age, level of education, name of university, and faculty. Help-seeking behavior was assessed by asking where the students usually seek help when having mental health problems. Mental health problems, how they cope with their stress and problems, and expectations of healthcare services when they need help were asked using the adaptation of tool developed by Mcluckie et al. to assess mental health literacy in young people [22]. This questionnaire was translated from English into Indonesian language, and the content has already been agreed upon in an expert consensus, consisting of child and adolescent psychiatrists, a pediatrician, and a public health doctor. In every subtopic, there were several lists of options, answered with “Yes”, “No”, or “Not applicable”. They were also allowed to add more options if they were not listed already.

### 2.3. Statistical Analysis

The demographic data, help-seeking behavior, and expectations of healthcare services were presented and summarized using descriptive analysis. For the bivariate analysis, Pearson chi-square tests or Fisher’s exact tests were used to compare the differences of mental health problems and coping mechanisms between students categorized by age (< 20 and ≥ 20 years old) and by university faculties (medical or nonmedical). We considered *p* values under 0.05 as significant. All analysis was done using the IBM SPSS Statistics version 24.

### 2.4. Ethics

Approval of the Ethical Committee of Medical Faculty Universitas Indonesia was obtained, with the protocol number 19-11-1354 on December 16, 2019. Participants were informed at the beginning of the survey that they could withdraw at any time by simply closing the browser page. All collected data were anonymous.

## 3. Results

### 3.1. Characteristics

Three hundred and ninety-three students from three different regions in Indonesia responded. Data were collected through online questionnaires. Students’ ages ranged between 16–24 years old, most of them were above 20 years old, thus currently enrolled in undergraduate study (95.1%), and more than half of the responders were medical students Table 1.

### 3.2. Help-Seeking Behavior

This part was assessed by asking students about who they think they or their peers would usually resort to when in need for help about their mental health problems Table 2. Most of the samples expressed that help is usually from friends (70.5%), psychologists (49.9%), and families (39.2%). Around a quarter of responders suggested health practitioners (23.7%) and counselors (21.9%) as sources of help. While only 18.8% referred to spiritual leaders, and a minority thought teachers (4.6%) and others (4.6%) could also be reached for help.

### 3.3. Mental Health Problems

The questionnaires explored mental health problems that university students usually dealt with. Further analysis includes comparison in ages (those who aged less than 20 years old or more) and faculties (medical and nonmedical students) Table 3. Furthermore, the problems are classified as problems related to internal factors and problems related to external factors. Almost all the responders answered that people of their age were having difficulty becoming more independent (92.1%) and managing their finances (91.3%). The proportion of students who faced problems dealing with more independent life was the same across ages during the transitional-age period (92%, *p* = 0.80) and from all faculty categories (92%, *p* = 0.93). Meanwhile, the percentage of financial management problems was higher in students aged more than 20 years rather than the younger age category (93% vs. 89%, *p* = 0.12). It is shown that medical students have financial problem more than nonmedical students (94% vs. 89%, *p* = 0.092) yet not significantly. Nearly four fifths of the students had problems with living expenses and education fees (79.1%). This proportion was similar between ages and faculties, ranging from 77–81% (*p* > 0.05).

During this transitional period, students also admitted that they had various academic-related problems. More than 90% of students had issues in their time management for daily schedules and exams preparation, stress control during exams, confidence to deal with academic or relationship challenges, learning difficulties, tasks, and academic burdens. Based on the subcategories analysis, the comparison of confidence problems between faculties (*p* = 0.02), learning challenges between ages (*p* = 0.05), and task and academic burdens between ages (*p* = 0.002) reached statistical significance.

Besides academic-related issues, students reported some problems in terms of their social life. Around 86% of students lived far from home and family, and 91% felt lonely. These proportions were similar to the further comparison between ages and faculties category. More than 80% of students had problems interacting with people inside or outside the campus and intimate relationships. The subgroup analysis showed similar percentages and did not differ significantly, except for one comparison: students younger than 20 years old were higher in number than their older counterparts (88% vs. 78%, *p* = 0.007). Around 88% of students in total or in the age or faculty categories ended a difficult relationship. Moreover, more than 70% of students experienced bullying, violence in a relationship, and sexuality problems. Among these problems, only the age comparison of students who had sexuality problems differed significantly. Sexuality problems happened in 71% of students younger than 20 years old and 78% in the older group (*p* = 0.096).

Ninety percent of students had issues in coming into a healthy way to solve problems. A little bit lower than this number, 86.3% of students had difficulties increasing their sense of well-being. However, an overwhelming number of students had problems in their emotional and stress management, which stood at 94.6% and 96.4%, respectively. Students who had issues in knowing when or where to go for physical and mental health problems ranged between 81% and 85%. About 84% of students did not fully understand the treatment for health problems. Likewise, approximately 92% of students could not help their friends or loved ones to overcome physical and mental health problems. The further comparison between ages and faculties also showed a similar figure and did not differ significantly (*p* > 0.05).

During this stressful period, students reported several psychiatric problems, which were alcohol or drug abuse (72.3%), game/internet addiction or gambling (77.4%), self-harm (75.3%), depression (88%), eating disorders (77.9%), and mood disorders and bipolar (88.3%). The most prevalent psychiatric problem was anxiety (95.4%). On the other hand, around 68% of students had severe psychiatric disorders. The proportion of students facing these problems in different age and faculty categories was similar to the mentioned figures and did not reach the statistical significance (*p* > 0.05).

### 3.4. Coping Mechanisms

The study also explored how people in transitional-age usually deal with difficulties Table 4. To cope with their stress, most of them chose to talk with their closest peers or someone they trusted (98.7%). All respondents aged older than 20 years prefer to speak to trusted persons, and 97% of students in the younger category had the same coping strategy (*p* = 0.02). Comparably, almost all subjects also chose to avoid the condition that triggers their stress (94.1%), while most of them had a healthy coping mechanism such as exercising (78.9%), practicing relaxation techniques (90.3%), doing their hobbies (96.4%), praying (95%), trying to find self-help strategies from the internet or books (89.8%), and writing in a journal or blog (54%).

The comparison of these coping categories between ages and faculties was similar in figure. However, only age comparison in exercise reached statistical difference (84% vs. 75% in the younger and older than 20 years old category, *p* = 0.03).

Some students seek help from other people. About 62.6% of students seek help from professional health or counselors, while approximately 90% of respondents discuss how to solve problems with someone who dealt with the same issues and copying the healthy coping strategies from the others. On the other hand, most students tried to solve their problem alone, by holding back and trying to forget (88.3%) and trying to deal with pressure (94.4%). Around 70% of respondents had uncontrolled anger and did nothing. Some students reported a change in their eating (88.5%) and sleeping (93%) pattern. The age and faculty analysis showed similar rates in each category, although each of these comparisons did not differ significantly (*p* > 0.05).

### 3.5. Expectations of Healthcare Services

This study also investigated what the youth expected of the healthcare services to make them more comfortable to seek help. Almost all of them answered that confidentiality was crucial (99.2%). Furthermore, it was also helpful to have a welcoming and friendly (99.2%) professionals who do not judge their condition (98.5%) and are open to listen to any of their problems (99%). Most respondents also suggested that a calm situation in a clinic or practice room made them more comfortable to attend the clinic or practice room (99.2%). In addition, it was found that continuous meeting with proper follow-up is important for their progress (96%) Table 5.

## 4. Discussion

Our study participant’s mean age was 19.94 ± 1.71 years old, with the age range of 16–24 years old. Most of the participants are students from the early years in their universities. More than half of them are medical students, which may be because transitional-age youth from health sciences background have greater enthusiasm and concern toward mental health research survey and are more responsive to answer the questionnaire; however, many are also from different faculties such as engineering, psychology, economy. According to Indonesian higher education statistics, students from health sciences, engineering, social humanities, and economics are about 50% of all enrolled students in Indonesia [23]. To decrease the impact of stigma and increase the freedom to express their opinions, the questionnaire asked about what their peers usually do or have. As an initial question, they were asked about their view of where their friends seek help when faced with stress. Most of their friends resorted to their peers, then followed by psychologists. Stigma about mental health institutions is still high in Indonesia, causing people to delay seeking help if they have problems related to mental health [24]. Like many other countries, the stigma is related to the bad image of the patients with mental illness, which is shown by the media, including the victimization [25].

Subjects were asked several questions divided into three categories: transition-age youth phase, coping mechanism, and expectation for healthcare system. Most of the respondents answered “yes“ to having problems in their transition-age, such as depression (88%) and anxiety (95.4%). In the subgroup analyses, the number of respondents in the younger or older than 20 years categories with depression and anxiety was similar. A previous study found that students older than 21 years were statistically related to major depressive disorders (OR = 1.959, 95% CI 1.3–2.9) [26]. In terms of faculty analyses, the proportion of depression among medical students was slightly higher than the nonmedical students (90% vs. 86%), while the rate of anxiety was almost the same in the two groups. Previous studies that investigated this factor of these psychiatric problems varied in the result. A study showed that perceived stress was slightly more common among medical students than nonmedical students (88.9% vs. 83.5%), while other research reported nonmedical students accounted for a larger percentage of depressive symptoms than their medical counterparts (59% vs. 48.4%) [27,28]. Despite the differences, the high prevalence of depression and anxiety among students in our result and this study agreed with the past studies, which showed depression and anxiety were higher among students than the general population [29,30,31]. A study found that perfectionism and impostor phenomenon, which happens when students continuously doubt their ability and fear that someone discover that they have intellectual incapability, were the factors that contribute to depression and anxiety among students in health sectors [32].

Other than that, 72.3% of the respondents reported substance and alcohol abuse as problems in our study. Further analysis showed a greater proportion of this issue occurred among students older than 20 years than the younger category (73% vs. 71%) and nonmedical than medical students (74% vs. 70%). This is similar to a prior study which stated that the prevalence of young adults who had used alcohol was 90%, while 61% of them reported having used drugs before. This report also mentioned that this substance addiction pattern increases with age, that 21% of them are 18–25 years old. Another study supported our result that alcohol abuse was more common among business students than medical students [33]. Eating disorders also occurred in a large portion of our study participants (77.9%). This finding is generally in agreement with a 10 year longitudinal study that found the prevalence of disordered eating behaviors increased from adolescence to young adulthood [34].

Academic stressors, such as time management for daily schedules, time management for exams preparation, stress control during exams, learning difficulties, task and academic burdens, were reported by over 90% of respondents in this transition age. Consistent with prior study, academic reason was the most common stressor among medical and nonmedical students [27]. Besides academic stressors, around 86% of our respondents had an issue because they were far from home and family. This finding was similar to studies that mentioned attending university for the first time can be challenging because of psychosocial adjustments or living away from their family [35,36].

The problem with sexuality also is one of the ubiquitous issues for transitional-age youth. This study found that this problem made up for almost 75% of the population study. A prior study by SAMSHA mentioned that sexuality contributed to 78% of the study population’s problem [37]. Studies on LGBT homeless youth showed varying results, but it suggested that around 20–40% of the worldwide youth identified themselves as sexual gender minorities. A study added by saying that lesbian, gay, bisexual, transgender, and other minor sexual groups have a higher risk of experiencing bullying and victimization compared to the heterosexual and cisgender community, which leads to increased cases of depression and suicide with more than 40% of them considering suicide. This is because they feel insecure about the future and have no support for their choices, get negative responses from others, and are unsure about themselves. Thus, they feel lonely and fearful [38].

Male and female perceptions about their youth life differ to some extent. Generally, there are no significant differences in the preparedness level between males and females. Most males felt they were prepared to deal with relationships but somehow not prepared for community involvement, while women feel more prepared about physical health but unprepared about independent living [39]. Preparedness for achieving higher education was rated the lowest [40]. Some studies stated that many college students are stressed because of increased demands and pressure from school and work overload, which leads to a negative impact on wellbeing. Thus, it is essential to have good self-regulation, proper coping mechanisms, emphasizing the individual’s active role, and good resilience [41].

In terms of coping with mental health problems, most of them answered that their friends mostly go to their closest peers to talk about their problems, followed by doing hobbies and praying. Carver et al. [42] defined a model for coping mechanisms as an effort from individuals to manage demands exceeding their resources. Albeit the complexity of coping mechanisms, there are three general coping forms during this transition period investigated in Carter’s study, which includes problem-focused coping, social support-seeking, and meaning-focused coping. In problem-focused coping, there are efforts to solve the problems, such as those that require planning, behavior coordination, and instrumental actions. This coping mechanism is found in people who use a problem-solving strategy to deal with general problems or self-identified stressors. The second form is social support-seeking. This form is multidimensional, includes emotional, instrumental, and informational support [43]. Some studies also mentioned that there are changes in an individual’s transition period toward increased communication skills, social support, and intergenerational openness. There is also a shift in source of support, from adults to peers. In meaning-focused coping, this form is to lower the negative impacts from the problems that cannot be solved, thus it is assumed to be palliative mechanisms, such as positive reappraisal, acceptance, and support the willingness to see the problem from a different point of view. This comes in agreement with the finding in this study where most participants from each age or faculty category chose to go to their peers for support, tried to find self-help strategies, and had discussion with others who faced similar situations. Supporting this, medical students used active ways of coping, such as planful problem-solving, seeking social support, and positive reappraisal [43,44]. Another study that involved nonmedical backgrounds showed that most students used the combination of appraisal-focused and problem-focused coping strategies [45]. On the other hand, most medical students from prior studies used recreational or religious activities to cope [44,46]. Unfortunately, many of the youth do not have the positive coping strategies and never had independently managed their health with the professionals. A study by Leipold et al. in 2018 [47] found that transitional-age youth usually terminate their appointment with the medical profession in the eighth meeting. Another study revealed that 18.6% of students probably did not seek help for an emotional problem [26].

Despite most of our respondents having healthy coping strategies, around half of them reported self-harming and having suicidal thoughts as coping strategies regardless of their age or faculty category. In comparison, the previous study found that the lifetime prevalence of nonsuicidal self-harm and suicidal thoughts among 18–23 years students was 20% and 19.8%, respectively. The proportion of these problems was more common among students living alone and having economic difficulty [48].

In this study, we also tried to analyze the expectations they have from the mental healthcare system to be more comfortable seeking help. Most of them agreed that confidentiality, open attitude, and friendliness from the medical professionals are the most important aspects. This is also similar to the study by Forouzan et al. in 2013, where they mentioned that they found the main domain that people expected in mental health professionals are good interrelationship with the patient. The patients said that they wanted to feel connected with the doctors, thus having a better bond and understanding. Healthcare accessibility is also mentioned to be the second most crucial domain, such as being located in large cities and having good home care and community mental health centers, which is also similar to our study. Continuity of care is also important because the participant in Forouzan et al. study mentioned that it is uncomfortable for them to change doctors and feel better if their doctor continues following them up after their meetings [49].

## 5. Limitations

Despite trying to avoid the impact of stigma on response by asking about their friend’s experiences, it is still possible that some answers are influenced by cultural views of mental illness or mental health problems. Although the minimal sample size was calculated to be 384 and the survey was gathered from three different zones of Indonesia, we would recommend gathering data from a bigger number with more equally distributed study background in the future.

## 6. Conclusions

Transitional-age youth is a golden period to transform and mature into adulthood. Challenges and new experiences are expected during this phase, along with higher pressures from new environments. Facing different challenges and depending on their coping mechanisms and resilience levels, some of the youth will be trapped into this phase’s pitfalls. It is important to analyze the possible challenges and stressors they experienced, how they cope with their problems, and their expectation from the healthcare where they promptly seek help. This effort may help them go through the phase successfully and mature into fully functioning adults. This study highlighted the most common issues the youth experienced. We found that while some of them already know how to deal with their problems, not all participants have adequate coping strategies. We also explored the expectations they have for healthcare to come up with recommendations to have a specific program to equip early years university students to improve the current condition, hoping that it will help them go through this phase. Universities could adopt a holistic public health approach to not only treat but also prevent and promote the mental health of their students. Web-based screening intervention or integrated mental health curriculum to improve mental health literacy could be beneficial. Moreover, universities could educate staffs and faculty personnel on early identification of mental health problems and pathways of referral to mental health professionals. Additionally, training students to offer support to their peers could also be one of the options to help students identify mental health problems and seek help.

## Figures and Tables

**Table 1 ijerph-18-04046-t001:** Sociodemographic characteristics of the sample.

Characteristics	n (%)
Age (in years)	
- 16	7 (1.8)
- 17	4 (1)
- 18	63 (16)
- 19	108 (27.5)
- 20	83 (21.1)
- 21	58 (14.8)
- 22	32 (8.2)
- 23	22 (5.6)
- 24	16 (4.1)
Current Education	
- Undergraduate	328 (95.1)
- Associate degree	7 (2)
- Other	58 (2.9)
Regions	
- West (Sumatra, Java)	269 (68.4)
- Center (Borneo, Sulawesi, Bali, Nusa Tenggara)	60 (15.3)
- East (Maluku, North Maluku, Papua)	64 (16.3)
Faculty	
- Medicine	202 (51.4)
- Engineering	28 (7.1)
- Psychology	13 (3.3)
- Economy and business	11 (2.8)
- Nursing	11 (2.8)
- Design	8 (2)
- Law	7 (1.8)
- Language and literature	7 (1.8)
- Public health	7 (1.8)
- Vocational study	4 (1)
- Pharmacy	3 (0.8)
- Agriculture	3 (0.8)
- Other (computer science, natural sciences, education, social sciences, politics, hospitality, tourism, cultural, and spiritual studies, etc.)	89 (22.6)
University	
- Public universities	244 (61.6)
- Private universities	88 (22.4)
- Other	61 (15.5)

**Table 2 ijerph-18-04046-t002:** Proportion of preferred help sources among students.

Seek Help for Mental Health Problems From:	n (%)
Friends	277 (70.5)
Psychologist	196 (49.9)
Family	154 (39.2)
Health practitioners	93 (23.7)
Counselor	86 (21.9)
Spiritual leader	74 (18.8)
Teachers	18 (4.6)
Others	18 (4.6)

**Table 3 ijerph-18-04046-t003:** Mental health problems among university students, in total, based on age and faculty.

Problems	Total		Age < 20 Years				Age ≥ 20 years				*p*-Value *	Medical Students				Non-Medical Students				*p*-Value *
	Yes	No	Yes		No		Yes		No			Yes		No		Yes		No		
	N (%)	N (%)	N	(%)	N	(%)	N	(%)	N	(%)		N	(%)	N	(%)	N	(%)	N	(%)	
Problems related to internal factors																				
Dealing with more independent life	362 (92.1)	31 (7.9)	167	(92)	15	(8)	195	(92)	16	(8)	0.80	178	(92)	15	(8)	184	(92)	16	(8)	0.93
Managing finances independently	359 (91.3)	34 (8.7)	162	(89)	20	(11)	197	(93)	14	(7)	0.12	181	(94)	12	(6)	178	(89)	22	(11)	0.09
Time management for daily schedules	367 (93.4)	26 (6.6)	174	(96)	8	(4)	193	(91)	18	(9)	0.10	183	(95)	10	(5)	184	(92)	16	(8)	0.26
Time management for exams preparation	370 (94.1)	23 (5.9)	174	(96)	8	(4)	196	(93)	15	(7)	0.25	182	(94)	11	(6)	188	(94)	12	(6)	0.89
Stress control during exams	366 (93.2)	27 (6.8)	171	(94)	11	(6)	195	(92)	16	(8)	0.54	183	(95)	10	(5)	183	(92)	17	(9)	0.19
Feeling confidence to deal with academic or relationship challenges	378 (96.2)	15 (3.8)	178	(98)	4	(2)	200	(95)	11	(5)	0.12	190	(98)	3	(2)	188	(94)	12	(6)	**0.02**
Learning difficulties	360 (91.6)	33 (8.4)	172	(95)	10	(5)	188	(89)	23	(11)	0.05	178	(92)	15	(8)	182	(91)	18	(9)	0.66
Feeling lonely	358 (91)	35 (9)	168	(92)	14	(8)	190	(90)	21	(10)	0.43	177	(92)	16	(8)	181	(91)	19	(9)	0.67
Sexuality	294 (74.8)	99 (5.2)	129	(71)	53	(29)	165	(78)	46	(22)	0.096	141	(73)	52	(27)	153	(77)	47	(23)	0.43
Come into a healthy way to solve problems	354 (90)	39 (10)	167	(92)	15	(8)	187	(89)	24	(11)	0.30	178	(92)	15	(8)	176	(88)	24	(12)	0.16
Increase the sense of well-being	339 (86.3)	54 (13.7)	154	(85)	28	(15)	185	(88)	26	(12)	0.37	166	(86)	27	(14)	173	(87)	27	(13)	0.88
Emotional management	372 (94.6)	21 (5.4)	174	(96)	8	(4)	198	(94)	13	(6)	0.43	187	(97)	6	(3)	185	(93)	15	(7)	**0.05**
Stress management	379 (96.4)	14 (3.6)	175	(96)	7	(4)	204	(97)	7	(3)	0.77	188	(97)	5	(3)	191	(96)	9	(4)	0.31
Alcohol or drug abuse	284 (72.3)	109 (27.7)	129	(71)	53	(29)	155	(73)	56	(27)	0.56	136	(70)	57	(30)	148	(74)	52	(26)	0.43
Game/internet addiction or gambling	304 (77.4)	89 (21.6)	138	(76)	44	(24)	166	(79)	45	(21)	0.50	149	(77)	44	(23)	155	(78)	45	(22)	0.99
Self-harm	296 (75.3)	97 (24.7)	138	(76)	44	(24)	158	(75)	53	(25)	0.98	140	(73)	53	(27)	156	(78)	44	(22)	0.21
Depression	346 (88)	47 (12)	160	(88)	22	(12)	186	(88)	25	(12)	0.94	174	(90)	19	(10)	172	(86)	28	(14)	0.20
Anxiety	375 (95.4)	18 (4.6)	173	(95)	9	(5)	202	(96)	9	(4)	0.74	183	(95)	10	(5)	192	(96)	8	(4)	0.57
Eating disorders	306 (77.9)	87 (22.1)	143	(79)	39	(21)	163	(77)	48	(23)	0.75	155	(80)	38	(20)	151	(76)	49	(24)	0.25
Mood disorders, bipolar	347 (88.3)	46 (11.7)	161	(88)	21	(12)	186	(88)	25	(12)	0.92	172	(89)	21	(11)	175	(88)	25	(12)	0.62
Severe psychiatric disorders	268 (68.2)	125 (31.8)	124	(68)	58	(32)	144	(68)	67	(32)	0.98	132	(68)	61	(32)	136	(68)	64	(32)	0.93
**Problems related to external factors**																				
Living expenses and education fees	311 (79.1)	82 (20.9)	141	(77)	41	(23)	170	(81)	41	(19)	0.45	154	(80)	39	(20)	157	(79)	43	(22)	0.75
Task and academic burdens	375 (95.4)	18 (4.6)	180	(99)	2	(1)	195	(92)	16	(8)	**0.002**	186	(96)	7	(4)	189	(95)	11	(5)	0.37
Far from home and family	338 (86)	55(14)	154	(85)	28	(15)	184	(87)	27	(13)	0.46	171	(89)	22	(11)	167	(84)	33	(16)	0.14
Interaction with peers in the campus	341 (86.8)	52 (13.2)	166	(91)	16	(9)	175	(83)	36	(17)	0.016	164	(85)	29	(15)	177	(89)	23	(11)	0.30
Interaction with people outside campus	326 (82)	67 (18)	161	(88)	21	(12)	165	(78)	46	(22)	**0.007**	160	(83)	33	(17)	166	(83)	34	(17)	0.97
Intimate relationship	338 (86)	55 (14)	152	(84)	30	(16)	186	(88)	25	(12)	0.18	162	(84)	31	(16)	176	(88)	24	(12)	0.25
End a difficult relationship	347 (88.3)	46 (11.7)	163	(89)	20	(11)	184	(88)	26	(12)	0.46	173	(90)	20	(10)	174	(87)	26	(13)	0.41
Bullying	295 (75)	98 (25)	142	(78)	40	(22)	153	(73)	58	(27)	0.2	142	(74)	51	(26)	153	(77)	47	(23)	0.50
Violence in a relationship	287 (73)	106 (27)	132	(73)	50	(27)	155	(73)	56	(27)	0.83	137	(71)	56	(29)	150	(75)	50	(25)	0.37
When to seek help for physical health problems	318 (81)	75 (19)	148	(81)	34	(19)	170	(81)	41	(19)	0.85	156	(81)	37	(19)	162	(81)	38	(19)	0.96
Where to seek help for physical health problems	323 (82.2)	70 (17.8)	150	(82)	32	(18)	173	(82)	38	(18)	0.91	158	(82)	35	(18)	165	(83)	35	(17)	0.87
When to seek help for mental health problems	332 (84.5)	61 (15.5)	154	(85)	28	(15)	178	(84)	33	(16)	0.94	163	(84)	30	(16)	169	(85)	31	(15)	0.99
Where to seek help for mental health problems	331 (84.2)	62 (15.8)	155	(85)	27	(15)	176	(83)	35	(17)	0.63	163	(84)	30	(16)	168	(84)	32	(16)	0.90
Treatment for health problems	331 (84.2)	62 (15.8)	152	(84)	30	(16)	179	(85)	32	(15)	0.72	163	(84)	30	(16)	168	(84)	32	(16)	0.90
Helping friends or loved ones to overcome physical and mental health problems	364 (92.6)	29 (7.4)	167	(92)	15	(8)	197	(93)	14	(7)	0.54	181	(94)	12	(6)	183	(92)	17	(9)	0.38

* Pearson chi-square.

**Table 4 ijerph-18-04046-t004:** Coping mechanisms of university students in total, based on age and faculty.

Coping Strategy to Deal with Mental Health Problems	Total		Age < 20 Years				Age ≥ 20 Years				*p*-Value *	Medical				Non-Medical				*p*-Value *
	Yes	No	Yes		No		Yes		No			Yes		No		Yes		No		
	N (%)	N (%)	N	(%)	N	(%)	N	(%)	N	(%)		N	(%)	N	(%)	N	(%)	N	(%)	
Talking to trusted person	388 (98.7)	5 (0.3)	177	(97)	5	(3)	211	(100)	0	(0)	0.02 **	190	(98)	3	(2)	198	(99)	2	(1)	0.68 **
Avoiding the events that create pressure at the time	370 (94.1)	23 (5.9)	167	(92)	15	(8)	203	(96)	8	(4)	0.06	180	(93)	13	(7)	190	(95)	10	(5)	0.46
Self-harm	202 (51.4)	191 (48.6)	99	(54)	83	(46)	103	(49)	108	(51)	0.27	94	(49)	99	(51)	108	(54)	92	(46)	0.29
Relieving stress to the nearest object	276 (70.2)	117 (29.8)	128	(70)	54	(30)	148	(70)	63	(30)	0.96	140	(73)	53	(27)	136	(68)	64	(32)	0.32
Seeking help to professional health or counselor	246 (62.6)	147 (37.4)	105	(58)	77	(42)	141	(67)	70	(33)	0.06	122	(63)	71	(37)	124	(62)	76	(38)	0.80
Exercise	310 (78.9)	83 (22.1)	152	(84)	30	(16)	158	(75)	53	(25)	0.03	156	(81)	37	(19)	154	(77)	46	(23)	0.35
Relaxation	355 (90.3)	38 (9.7)	165	(91)	17	(9)	190	(90)	21	(10)	0.83	176	(91)	17	(9)	179	(90)	21	(11)	0.57
Doing a hobby or interests	379 (96.4)	14 (3.6)	177	(97)	5	(3)	202	(96)	9	(4)	0.41	186	(96)	7	(4)	193	(97)	7	(4)	0.95
Holding back and trying to forget	347 (88.3)	46 (11.7)	159	(87)	23	(13)	188	(89)	23	(11)	0.59	176	(91)	17	(9)	171	(86)	29	(15)	0.08
Praying	373 (95)	20 (5)	177	(97)	5	(3)	196	(93)	15	(7)	0.05	186	(96)	7	(4)	187	(94)	13	(7)	0.19
Going to the doctor more often for physical health problem	176 (44.8)	217 (55.2)	81	(45)	101	(55)	95	(45)	116	(55)	0.91	88	(46)	105	(54)	88	(44)	112	(56)	0.75
Desperate and having suicidal thought	227 (57.8)	166 (42.2)	103	(57)	79	(43)	124	(59)	87	(41)	0.66	109	(56)	84	(44)	118	(59)	82	(41)	0.61
Trying to find self-help strategies from internet or books	353 (89.8)	40 (10.2)	166	(91)	16	(9)	187	(89)	24	(11)	0.39	177	(92)	16	(8)	176	(88)	24	(12)	0.22
Having discussions on how to solve problems with someone who dealt with the same problems	354 (90)	39 (10)	164	(90)	18	(10)	190	(90)	21	(10)	0.98	174	(90)	19	(10)	180	(90)	20	(10)	0.96
Copying other friends or seniors that have healthy problem solving	344 (87.5)	49 (12.5)	156	(86)	26	(14)	188	(89)	23	(11)	0.31	172	(89)	21	(11)	172	(86)	28	(14)	0.35
Eating more or less	348 (88.5)	45 (11.5)	161	(88)	21	(12)	187	(89)	24	(11)	0.95	172	(89)	21	(11)	176	(88)	24	(12)	0.72
Sleeping more or less	365 (93)	28 (7)	168	(92)	14	(8)	197	(93)	14	(7)	0.68	176	(91)	17	(9)	189	(95)	11	(6)	0.20
Trying to deal with pressure	371 (94.4)	22 (5.6)	174	(96)	8	(4)	197	(93)	14	(7)	0.33	181	(94)	12	(6)	190	(95)	10	(5)	0.60
Uncontrolled anger	279 (71)	114 (29)	127	(70)	55	(30)	152	(72)	59	(28)	0.62	138	(72)	55	(28)	141	(71)	59	(30)	0.83
Surrendering and doing nothing	294 (74.8)	99 (25.2)	136	(75)	46	(25)	158	(75)	53	(25)	0.97	149	(77)	44	(23)	145	(73)	55	(28)	0.28
Writing in journal or blog	212 (54)	181 (46)	93	(51)	89	(49)	119	(56)	92	(44)	0.29	107	(55)	86	(45)	105	(53)	95	(48)	0.56

* Pearson chi-square, ** exception, used Fisher’s exact test.

**Table 5 ijerph-18-04046-t005:** Youth expectations of the healthcare services.

Questions	Answers	
	Yes	No
	N (%)	N (%)
Confidentiality	390 (99.2)	3 (0.8)
Nonjudgmental	387 (98.5)	6 (1.5)
Welcoming and friendly	390 (99.2)	3 (0.8)
Open attitude of health workers	389 (99)	4 (1)
Calm situation	390 (99.2)	3 (0.8)
The atmosphere describes service for young people	363 (92.4)	30 (7.6)
Inside the university		
	226 (57.5)	167 (42.5)
Near home	310 (78.9)	83 (21.1)
Acknowledge by family	175 (44.5)	218 (55.5)
Affordable	391 (99.5)	2 (0.5)
Insurance friendly	370 (94.2)	23 (5.8)
Can be accessed online	332 (84.5)	61 (15.5)
Face to face meeting	376 (95.7)	17 (4.3)
Take place regularly or continuously in a certain period of time	377 (96)	16 (4)
Group therapy	203 (51.7)	190 (48.3)
Using audiovisual	342 (87)	51 (13)
Material for guidance to be done at home	358 (91)	35 (9)

## Data Availability

The data presented in this study are available on request from the corresponding author. The data are not publicly available due to privacy reason.

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
