# Peer review of "Mental Health Problems and Needs among Transitional-Age Youth in Indonesia"

_ijerph, 2021, doi:10.3390/ijerph18084046_

Round 1

Reviewer 1 Report

The content of the article is consistent with the stated objetive. In my perspective it present us the results are very useful for future actions and derives many of the actions that we must carry out in the community.  e.g. The 80% of students had problems interacting with people inside and outside the campus... About 87% of students did not understand well about the treatment for health problems and more than 70% of students experienced bullying, violence in a relationship and sexuality problems. 

However, and with the purpose of improving the presentation of the text, I would like to share two recommendation for your consideration:

  1. The section 3.4 Coping Mechanisms. It would be convenient to eliminate the last two ideas, ( 250-251)which are already mentioned in another part of the article (380-385). It seems to me that these ideas are not in relation to the previous data. 
  2. is it possible to reclassify the list of problems presented in the table 3?  For example those related with the person himself and problems of the person and their environment  

Reviewer 2 Report

The work provides an important study linked to young people's mental health and well-being and novel study in the context of Indonesia.

Research design / methodology

The data has a large majority of respondents  form medical schools- this might need to be discussed in limitations of study and as a context in methodology. Might this have impacted on the overall data statistically. Are there more medical students in Indonesia than other types of students to account for the high number in your data, or are they simply a group more likely to respond to the questionnaire. This needs to be considered.

Your conclusions could provide more detailed recommendations in terms of developing mechanisms to support mental health and well-being. Could you lend from work in other countries which have mental health services within Universities and peer support networks etc.

Some proof reading required for some weak sentences

For example,

Line 53

Psychologically, in this transitional phase, while people transform into mature and 
responsible adults, things could become askew. This sentence should be reworded 'askew ' is wrong context.

Reword

Psychologically, in this transitional phase, things could become challenging for a young person and they could face difficulties as they develop which can impact on heir mental health well-being

Line 120

done by  Zeng et al.  would be better as research undertaken by Zeng et al. discovered that the pooled

Line 250

dismally 51.4% and 57.8% of students thought of self-harming and having suicidal thought when faced with a stressful event.  

Dismally is too emotive use 'significantly' instead

Line 272

The calm situation in the clinic or practice room made them more comfortable to come (99.2%)

This could be rewritten as

99.2% of respondents suggested a calm situation in a clinic or practice room made them more comfortable to attend the clinic or practice room (99.2%).

I suggest you read through the work and edit carefully so flow of writing is maintained throughout
